# Genomic surveillance of SARS-CoV-2 Omicron variants on a university campus

Ana A. Weil ⬡ [1] ✉, Kyle G. Luiten ⬡ [1], Amanda M. Casto ⬡ [1,2], Julia C. Bennett ⬡ [1], Jessica O'Hanlon[1], Peter D. Han[3,4], Luis S. Gamboa[3,4], Evan McDermot[3], Melissa Truong[3], Geoffrey S. Gottlieb[1,5,6], Zack Acker[3], Caitlin R. Wolf[1], Ariana Magedson[1], Eric J. Chow ⬡ [1], Natalie K. Lo[1], Lincoln C. Pothan[3,6], Devon McDonald[1], Tessa C. Wright[1], Kathryn M. McCaffrey[3], Marlin D. Figgins[2], Janet A. Englund[7], Michael Boeckh[2], Christina M. Lockwood[3,8], Deborah A. Nickerson[4], Jay Shendure ⬡ [3,4,9], Trevor Bedford[2,3,4,9], James P. Hughes[10], Lea M. Starita[3,4] & Helen Y. Chu[1]

Novel variants continue to emerge in the SARS-CoV-2 pandemic. University testing programs may provide timely epidemiologic and genomic surveillance data to inform public health responses. We conducted testing from September 2021 to February 2022 in a university population under vaccination and indoor mask mandates. A total of 3,048 of 24,393 individuals tested positive for SARS-CoV-2 by RT-PCR; whole genome sequencing identified 209 Delta and 1,730 Omicron genomes of the 1,939 total sequenced. Compared to Delta, Omicron had a shorter median serial interval between genetically identical, symptomatic infections within households (2 versus 6 days, $P = 0.021$). Omicron also demonstrated a greater peak reproductive number (2.4 versus 1.8), and a 1.07 (95% confidence interval: 0.58, 1.57; $P < 0.0001$) higher mean cycle threshold value. Despite near universal vaccination and stringent mitigation measures, Omicron rapidly displaced the Delta variant to become the predominant viral strain and led to a surge in cases in a university population.

Persistent SARS-CoV-2 circulation has led to the continued emergence of variants of concern (VOCs). On November 26, 2021, the World Health Organization designated Pango lineage B.1.1.529 as Omicron, a VOC which rapidly spread globally. Omicron is classified into sublineages BA.1, BA.2, and BA.3, etc., while BA.1 and BA.2 have several designated sublineages[1]. Mutations of the Omicron variant have demonstrated enhanced transmissibility despite widespread population immunity, as evidenced by the exponential increase in cases over shorter time periods compared to prior VOCs[2–4]. There is also population-level, genomic, and in vitro evidence of decreased

vaccine effectiveness against Omicron compared to the Delta variant and of partial evasion of vaccine-induced immunity by Omicron, leading to high numbers of breakthrough infections[5–8]. Studies have shown mixed results on differences in Omicron viral load compared to the Delta variant, with evidence of either lower or comparable viral loads for Omicron[9–15]. Omicron household transmission has been reported to have a higher attack rate and lower serial interval compared to Delta, although the majority of studies to date have not used genomic data to assess the serial intervals in intra-household transmission[16–22]. There remain gaps in our understanding of the

[1]Department of Medicine, University of Washington, Seattle, WA, USA. [2]Vaccine and Infectious Diseases Division, Fred Hutchinson Cancer Research Center, Seattle, WA, USA. [3]Brotman Baty Institute, Seattle, WA, USA. [4]Department of Genome Sciences, University of Washington, Seattle, WA, USA. [5]Environmental Health & Safety Department, University of Washington, Seattle, WA, USA. [6]Department of Global Health, University of Washington, Seattle, WA, USA. [7]Seattle Children's Research Institute, Department of Pediatrics, University of Washington, Seattle, WA, USA. [8]Department of Laboratory Medicine and Pathology, University of Washington, Seattle, WA, USA. [9]Howard Hughes Medical Institute, Seattle, WA, USA. [10]Department of Biostatistics, University of Washington, Seattle, WA, USA. ✉e-mail: anaweil@uw.edu

transmission dynamics and molecular epidemiology of VOC emergence in US populations.

Throughout the COVID-19 pandemic, university campuses have been sites of SARS-CoV-2 outbreaks[23–26]. Many universities provide free, convenient testing to facilitate SARS-CoV-2 surveillance within campus communities[23,25,27]. Using data collected from September 2021 to February 2022 through a campus testing program, we describe the rapid emergence of Omicron in a highly vaccinated university community and the clinical characteristics and transmission dynamics of the Omicron variant compared to the Delta variant. We used molecular epidemiology to track the emergence of variants and examine intra-esidence infections in congregate living settings.

## Results

A total of 37,985 participants were enrolled as of February 14, 2022. Seventy-four thousand nine hundred ninety-five samples were collected from 24,393 participants between September 10, 2021, and February 14, 2022. A total of 3630 samples (4.8%) were SARS-CoV-2 positive, representing 3048 individuals. Genomic sequencing of 2101 samples from 1939 individuals identified 209 Delta and 1730 Omicron cases (Fig. 1). Six individuals had both sequenced Delta and Omicron infections during the study period; only the first infection was considered for each individual.

### Clinical characteristics

The median age of participants with infection was 20 years (range 18–66) for Delta and 21 years (range 17–79) for Omicron (Table 1). Most SARS-CoV-2 cases were among students (90.9% of Delta cases, compared to 87.9% of Omicron cases). Residing in a household with a density of ≥6 was reported for 34.0% of Delta and 23.8% of Omicron cases. 18.2% of Delta and 18.3% of Omicron cases were asymptomatic at the time of swabbing. Among symptomatic cases, the most reported symptoms were rhinorrhea/congestion (69.6% and 62.4% for Delta and Omicron, respectively), cough (59.1% and 61.5%), and sore throat (56.1% and 69.2%). Loss of sense of taste or smell was more common among Delta cases (11.1% of those with Delta vs. 2.8% of those with Omicron, $P < 0.001$). Myalgias, fever, and chills were more prominent in Omicron cases (29.8%, 34.1%, and 24.4%, respectively) than Delta (15.8, 25.1, and 15.8; $P < 0.001$, $P = 0.025$, and $P = 0.015$). The mean time from symptom onset to the first positive sample was 2.82 days (standard deviation [SD]: 2.03) for Delta and 2.76 days (1.93) for Omicron.

COVID-19 vaccination status was known at the time of infection for 141 (67.5%) of Delta and 1182 (68.3%) of Omicron cases. For those with known vaccination status, 1147 (97.0%) with Delta and 137 (97.1%) with Omicron completed a primary series, with an additional 2 (1.4%) with Delta and 3 (0.3%) with Omicron that partially completed the primary series. Three (2.1%) with Delta and 337 (28.5%) with Omicron received a booster dose at least two weeks before infection, and 1 (0.7%) with Delta and 42 (3.6%) with Omicron received a booster dose less than two weeks before infection. Two (1.4%) with Delta and 32 (2.7%) with Omicron were unvaccinated. Intervals between infection and last mRNA vaccine dose received are shown by vaccination status and variant in Fig. 2. Most vaccinated participants completed their primary series by early Spring 2021, and the number of days since primary series for Omicron cases (median 271 days, IQR: 251, 292) were higher than for Delta cases (median 194 days, IQR: 169, 224).

In our Ct value analysis, we compared the first positive, sequenced sample from each individual detected using our standard swab type (RHINOstic$_{TM}$ swabs) ($N = 1870$, excluding 27 Delta and 42 Omicron cases detected using US Cotton #3 swabs). Adjusting for age, symptom status, and average RNase P gene value, the mean Orf1b Ct was 1.07 higher (95% confidence interval 0.58, 1.57; $P < 0.00001$) among Omicron compared to Delta cases. Mean adjusted difference in Orf1b Ct comparing symptomatic to asymptomatic cases was −1.11 (95% CI,

−1.50, −0.74; $P < 0.00001$) and for each 1-unit increase in average RNase P gene value was 0.39 (95% CI, 0.35, 0.43; $P < 0.00001$). Results did not change in a sensitivity analysis without adjustment for symptom status (mean Ct 1.07 [0.57, 1.57] higher among Omicron compared to Delta and mean increase of 0.38 [0.34, 0.43] for each 1-unit increase in average RNase P gene value). Among symptomatic individuals ($N = 1466$), days since symptom onset was significantly associated with a higher Ct value (0.29 higher per day, [95% CI: 0.20, 0.38], $P < 0.00001$) and therefore, lower semiquantitative viral loads were observed in those with a longer duration of symptoms at the time of sample collection (Table 2). We did not find a difference in semiquantitative viral load comparing Omicron Pango lineages BA.1 and BA.2 ($N = 1688$, Supplemental Table 1).

### Intraresidence transmission

Among the 1939 SARS-CoV-2 genomes, we identified 13 residences with multiple sequenced Delta cases and 136 residences with multiple sequenced Omicron cases. Phylogenetic and pairwise distance analyses of these genomes indicated that many cases within the same residence were likely the result of more than one introduction event. Thus, we restricted the analysis to 78 clusters, including 173 individuals with identical viral genomes within the same residence ($N = 25$ residents for Delta, and 148 for Omicron). Thirty individuals reported that symptoms began on the same day as another individual in the cluster, and 53 collected their first positive sample on the same day as another individual in the cluster. All identical viral genomes within a single household were detected within a maximum serial interval of 15 days. Forty-four clusters included more than one symptomatic individual and more than one unique symptom onset date. Among these clusters, the median serial interval between symptom onset of the index and a subsequent case was longer for 8 subsequent cases in 7 Delta clusters (median 6 days, range [1–10]) compared to for 43 subsequent cases in 37 Omicron clusters (median 2 days, [1–9]) ($P = 0.021$, Supplemental Fig. 1).

### Genomic analysis

A phylogenetic tree shown in Fig. 3A includes all 209 Delta genomes shown with 1174 randomly selected genomes from samples collected in Washington state over the same time period. A phylogenetic tree containing all 1939 sequenced viral genomes is shown in Fig. 3B, illustrating the rapid replacement of Delta by Omicron on the university campus in December 2021. Three monophyletic clusters containing exclusively or almost exclusively study genomes ($N = 35, 24, 66$ total genomes and $N = 35, 23, 66$ HCT genomes) are boxed in Fig. 3A; -60% of all study Delta genomes fall into one of these three groups. Bootstrap values for all 3 Delta clusters were 100%. The maximum pairwise distance between two studies Delta samples was 60 nucleotide differences, and the average distance was 18.54. One hundred sixteen (56%) of these samples were genetically identical to at least one other study sample. Supplemental Table 2 includes additional information about the demographic information of Delta clusters. The tree in Fig. 3C includes all 1730 Omicron genomes with 1512 randomly selected genomes from samples collected in Washington state over the same time period. Relative to the Delta genomes, the study Omicron genomes are more evenly distributed throughout the tree, particularly genomes from samples collected in January and February. Bootstrap values were also, on average, much lower for nodes in the Omicron tree relative to the Delta tree (average bootstrap value 32.5% versus 66.8%). The maximum pairwise distance between two studies Omicron samples was 89, and the average distance was 7.10 (72 and 6.01, respectively, excluding BA.2 samples). One thousand thirty-nine (60%) of Omicron samples were genetically identical to at least one other study sample. Among 1730 sequenced Omicron samples, 24 were of the BA.2 lineage. The maximum pairwise distance among these was 9, and 19 (79%) were identical to at least one other study genome.

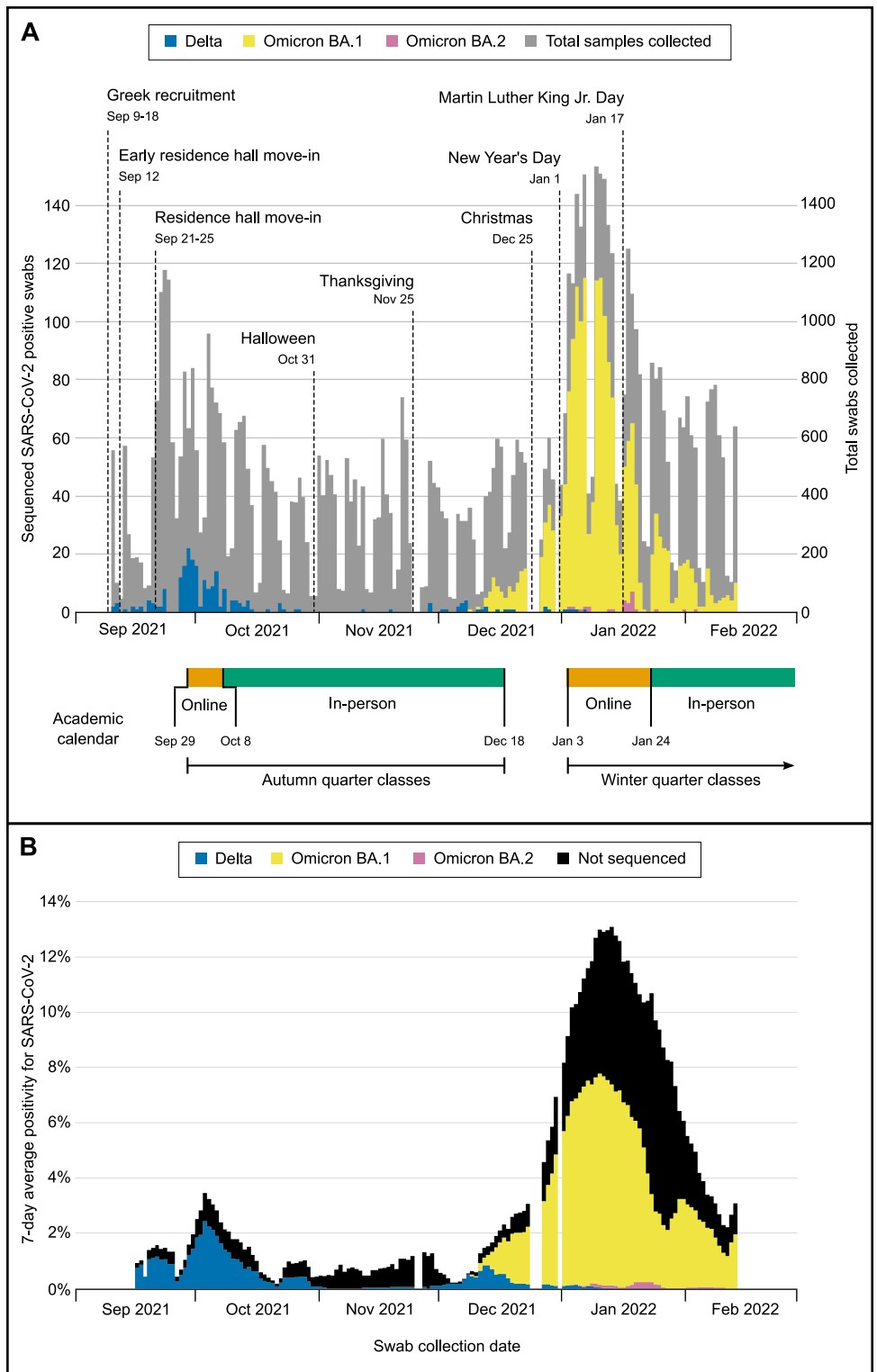

**Fig. 1 | Sequenced SARS-CoV-2-positive samples collected from September 10, 2021 to February 14, 2022, by Pango Lineage. A** Daily counts of total samples collected and positive samples. **B** SARS-CoV-2 7-day average percent positivity. Campus events, holidays, and breaks in coursework that impacted university populations are shown. Testing demand was reduced on weekends, and operations were paused for holidays (represented by gaps in testing), inclement weather, and campus closures.

To estimate the number of introduction events of Delta into the campus population needed to explain the sequenced samples, we created a phylogenetic tree including all sequenced Delta study samples and all publicly available genomes for Delta viruses collected in Washington state from September 1, 2021, to February 14, 2022, for a total of 209 genomes from our study and 15,406 Washington state genomes. By determining the likely classification of internal nodes as either campus or community-based, we estimated that the sequenced Delta samples resulted from 83 different introductions of the variant with 2.5 sequenced cases per introduction. We performed the same analysis for the sequenced Omicron samples using 14,359 publicly available Omicron genomes from samples collected in Washington

## Table 1 | Demographic characteristics and symptom profiles for Delta and Omicron variant infected study participants, September 10, 2021, to February 14, 2022

| | Delta (N = 209) | All Omicron (N = 1730) | Omicron BA.1 (N = 1706) | BA.2 (N = 24) |
|---|---|---|---|---|
| **Collection date range** | Sep 10, 2021–Jan 7, 2022 | Dec 09, 2021–Feb 14, 2022 | Dec 09, 2021–Feb 14, 2022 | Jan 3, 2022–Feb 4, 2022 |
| **Age (years), Median [Min, Max]** | 20 [18, 66] | 21 [17, 79] | 21 [17, 79] | 21 [18, 32] |
| **Sex, N (%)** | | | | |
| Male | 86 (41.1) | 715 (41.3) | 705 (41.3) | 10 (41.7) |
| Female | 122 (58.4) | 1007 (58.2) | 993 (58.2) | 14 (58.3) |
| Other | — | 1 (0.1) | 1 (0.1) | — |
| Prefer not to say | 1 (0.5) | 7 (0.4) | 7 (0.4) | — |
| **Comorbidities (one or more), N (%)** | 55 (26.3) | 445 (25.7) | 441 (25.9) | 4 (16.7) |
| **Race[a], N (%)** | | | | |
| American Indian or Alaska Native | 2 (1.0) | 10 (0.6) | 9 (0.5) | 1 (4.2) |
| Asian | 37 (17.7) | 598 (34.6) | 580 (34.0) | 18 (75.0) |
| Black | 2 (1.0) | 41 (2.4) | 40 (2.3) | 1 (4.2) |
| Native Hawaiian or other Pacific Islander | — | 4 (0.2) | 4 (0.2) | — |
| White | 145 (69.4) | 806 (46.6) | 803 (47.1) | 3 (12.5) |
| Other | 4 (1.9) | 85 (4.9) | 84 (4.9) | 1 (4.2) |
| Prefer not to say | 5 (2.4) | 48 (2.8) | 48 (2.8) | — |
| Multiple races[b] | 14 (6.7) | 138 (8.0) | 138 (8.1) | — |
| **Affiliation, N (%)** | | | | |
| Student | 190 (90.9) | 1520 (87.9) | 1497 (87.7) | 23 (95.8) |
| On-campus resident | 62 (32.6) | 414 (27.2) | 405 (27.1) | 9 (39.1) |
| Fraternity or sorority resident | 60 (31.6) | 205 (13.5) | 205 (13.8) | — |
| Staff | 12 (5.7) | 167 (9.7) | 166 (9.7) | 1 (4.2) |
| Faculty | 5 (2.4) | 37 (2.1) | 37 (2.2) | — |
| Other | 2 (1.0) | 6 (0.3) | 6 (0.4) | — |
| **Household density[c], N (%)** | | | | |
| 1 | 25 (12.0) | 206 (11.9) | 206 (12.1) | — |
| 2 | 62 (29.7) | 519 (30.0) | 508 (29.8) | 11 (45.8) |
| 3 | 21 (10.0) | 299 (17.3) | 295 (17.3) | 4 (16.7) |
| 4 | 24 (11.5) | 226 (13.1) | 219 (12.8) | 7 (29.2) |
| 5 | 6 (2.9) | 69 (4.0) | 69 (4.0) | — |
| 6 or more | 71 (34.0) | 411 (23.8) | 409 (24.0) | 2 (8.3) |
| Mean (SD)[d] | 3.66 (1.91) | 3.38 (1.75) | 3.39 (1.75) | 3.08 (1.25) |
| **Primary series, N (%)** | | | | |
| Primary series complete | 137 (65.6) | 1147 (66.3) | 1132 (66.4) | 15 (62.5) |
| BNT162b2 | 95 (69.3) | 694 (60.6) | 683 (60.4) | 11 (73.3) |
| mRNA-1273 | 29 (21.2) | 268 (23.4) | 267 (23.6) | 1 (6.7) |
| Ad26.COV2.S | 11 (8.0) | 49 (4.3) | 48 (4.2) | 1 (6.7) |
| ChAdOx1-S | 1 (0.7) | 14 (1.2) | 12 (1.1) | 2 (13.3) |
| Mix and match | 1 (0.7) | 4 (0.3) | 4 (0.4) | — |
| Unknown manufacturer and date | — | 117 (10.2) | 117 (10.3) | — |
| Days since primary series, Mean (SD) | 190 (40.4) | 270 (51.9) | 271 (51.7) | 241 (55.6) |
| Partially complete primary series | 2 (1.0) | 3 (0.2) | 3 (0.2) | — |
| Not vaccinated | 2 (1.0) | 32 (1.8) | 32 (1.9) | — |
| Invalid dates or no information reported | 68 (32.5) | 548 (31.7) | 539 (31.6) | 9 (37.5) |
| **Booster dose, N (%)** | | | | |
| Fully boosted | 3 (1.4) | 337 (19.5) | 333 (19.5) | 4 (16.7) |
| BNT162b2 | 1 (33.3) | 140 (41.5) | 137 (41.1) | 3 (75.0) |
| mRNA-1273 | — | 94 (27.9) | 94 (28.2) | — |
| Ad26.COV2.S | 2 (66.7) | 10 (3.0) | 9 (2.7) | 1 (25.0) |

## Table 1 (continued) | Demographic characteristics and symptom profiles for Delta and Omicron variant infected study participants, September 10, 2021, to February 14, 2022

| | Delta (N = 209) | All Omicron (N = 1730) | Omicron BA.1 (N = 1706) | BA.2 (N = 24) |
|---|---|---|---|---|
| Unknown manufacturer | — | 93 (27.6) | 93 (27.9) | — |
| Days since booster dose, Mean (SD) | 194 (60.4) | 58.4 (51.2) | 57.6 (49.3) | 108 (125) |
| Partially boosted | 1 (0.5) | 42 (2.4) | 42 (2.5) | — |
| Not boosted | 98 (46.9) | 752 (43.5) | 744 (43.6) | 8 (33.3) |
| Invalid dates or no information reported | 107 (51.2) | 599 (34.6) | 587 (34.4) | 12 (50.0) |
| **Symptom presence, N (%)** | | | | |
| Asymptomatic | 38 (18.2) | 316 (18.3) | 312 (18.3) | 4 (16.7) |
| Symptomatic | 171 (81.8) | 1414 (81.7) | 1394 (81.7) | 20 (83.3) |
| COVID-19-like illness[e] | 42 (24.6) | 436 (30.8) | 429 (30.8) | 7 (35.0) |
| Influenza-like illness[f] | 49 (28.7) | 511 (36.1) | 503 (36.1) | 8 (40.0) |
| Symptom duration (days)[g], Mean (SD) | 2.82 (2.03) | 2.76 (1.93) | 2.77 (1.94) | 2.11 (1.05) |
| Number of symptoms[h], Mean (SD) | 3.46 (2.37) | 4.17 (2.90) | 4.18 (2.90) | 3.35 (2.60) |
| Runny or stuffy nose | 119 (69.6) | 883 (62.4) | 871 (62.5) | 12 (60.0) |
| Cough | 101 (59.1) | 869 (61.5) | 857 (61.5) | 12 (60.0) |
| Sore throat or itchy/scratchy throat | 96 (56.1) | 979 (69.2) | 966 (69.3) | 13 (65.0) |
| Increased trouble with breathing | 8 (4.7) | 110 (7.8) | 109 (7.8) | 1 (5.0) |
| Muscle or body aches | 27 (15.8) | 422 (29.8) | 420 (30.1) | 2 (10.0) |
| Headache | 68 (39.8) | 667 (47.2) | 662 (47.5) | 5 (25.0) |
| Feeling feverish | 43 (25.1) | 482 (34.1) | 476 (34.1) | 6 (30.0) |
| Feeling more tired than usual | 44 (25.7) | 490 (34.7) | 484 (34.7) | 6 (30.0) |
| Chills or shivering | 27 (15.8) | 345 (24.4) | 339 (24.3) | 6 (30.0) |
| Sweats | 21 (12.3) | 242 (17.1) | 239 (17.1) | 3 (15.0) |
| Rash | 1 (0.6) | 15 (1.1) | 15 (1.1) | — |
| New loss of taste or smell | 19 (11.1) | 39 (2.8) | 39 (2.8) | — |
| Nausea or vomiting | 6 (3.5) | 143 (10.1) | 143 (10.3) | — |
| Ear pain or ear discharge | 6 (3.5) | 63 (4.5) | 62 (4.4) | 1 (5.0) |
| Eye pain | 2 (1.2) | 76 (5.4) | 76 (5.5) | — |
| Diarrhea | 3 (1.8) | 71 (5.0) | 71 (5.1) | — |

[a]Race is divided into mutually exclusive groups.
[b]Multiple races included participants reporting more than one of these groups.
[c]Household density was defined as the number of people sharing the same kitchen or living space.
[d]A household density of 6 was assumed for participants who reported more than 6 household members.
[e]COVID-19-like illness (CLI) was defined as self-reported fever, chills, and/or shivering, with cough and/or shortness of breath.
[f]Influenza-like illness (ILI) was defined as self-reported fever, chills, and/or shivering, with cough and/or sore throat.
[g]Duration between symptom onset and first SARS-CoV-2-positive result in symptomatic participants.
[h]Unique symptoms reported by a participant within 7 days before and after collecting their first SARS-CoV-2-positive swab.

state up to February 14, 2022. We estimate that 1021 introduction events were necessary to explain the 1730 sequenced Omicron cases, with 1.7 sequenced cases per introduction. We also assessed the Omicron BA.2 subvariant viruses separately. We created a tree containing the 24 BA.2 viral genomes generated from samples collected on campus plus 126 BA.2 genomes from samples collected in Washington up until February 14, 2022. We estimated that the 24 sequenced study cases resulted from 8 different introductions with 3.0 sequenced cases per introduction. To assess the accuracy of the Delta and Omicron introduction number estimates, we repeated these analyses using smaller pools of Washington state (non-study) genomes. This assessment showed that the estimate of Delta introduction events would be

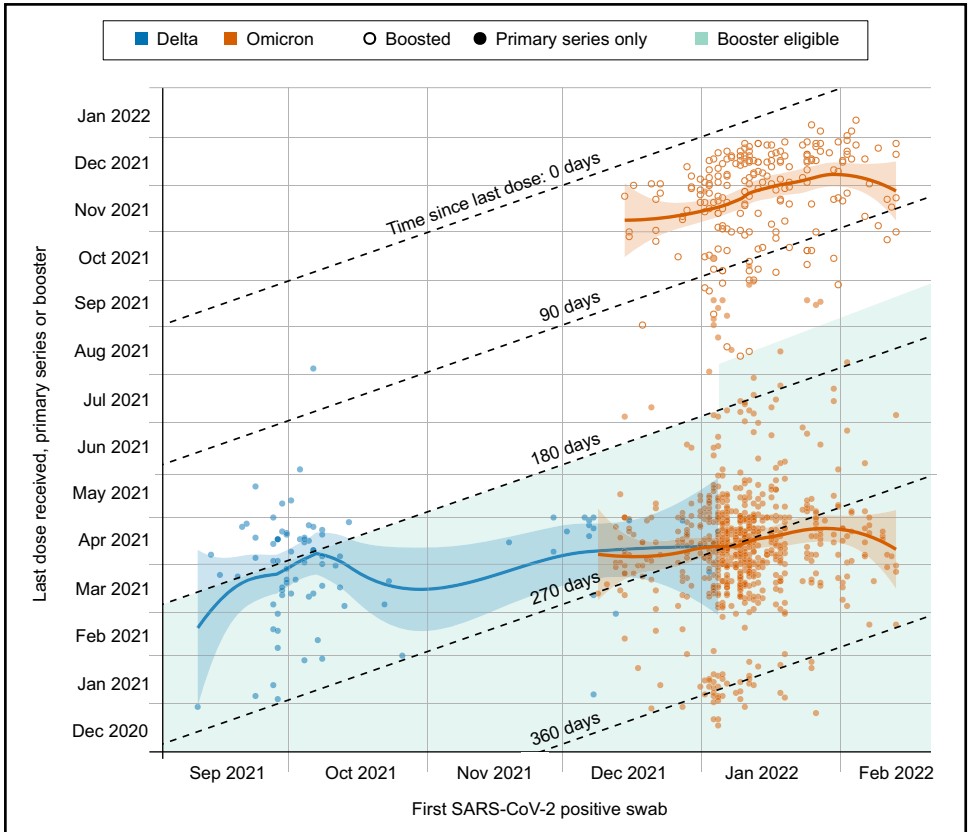

**Fig. 2 | Interval between Delta or Omicron infection and last mRNA vaccine dose received.** Local fitted polynomial regression using the loess function of the R stats package, with α = 0.75 and second-degree polynomials is shown as shaded 95% CIs shown for each variant and by booster status for Omicron. Period of booster eligibility is 180 or 150 days (beginning January 4, 2022) after primary series completion. In the US, a booster dose of BNT162b2 was available with limited eligibility on Sept 25, 2021, mRNA-1273 and Ad26.COV.2.S on Oct 20, 2021, and general eligibility on Nov 21, 2021. Not shown are participants who were unvaccinated, partially vaccinated, had unknown vaccination status, or received a vaccine other than BNT162b2 or mRNA-1273.

unlikely to change even if more Washington state genomes were available, though it was unclear if this was the case for the Omicron estimate (Supplemental Fig. 2).

### Transmission dynamics

To quantify the degree to which each variant impacted on-campus transmission rates, we estimated variant-specific transmission dynamics following previously established methods[28]. Here we find that the Rt associated with the September to October Delta outbreak peaked at 1.8 (95% credible intervals [CI] 1.3–2.4) and declined rapidly below 1, while the Rt associated with the December to January Omicron outbreak peaked at 2.4 (95% CI 1.9–2.8) and declined below 1 over a longer period (Fig. 4). These differences in Rt are reflected in the relative magnitudes of the September to October Delta outbreak compared to the December to January Omicron outbreak (Fig. 4).

### Discussion

In a large, urban university campus with widely available testing, stringent mitigation measures, and near-universal vaccination, the Omicron variant rapidly displaced the Delta variant to become the predominant viral strain over a two-week period. Fever, myalgia, and chills were more commonly reported in Omicron cases and loss of taste and smell in Delta cases. Ct values were, on average higher for Omicron cases. Using genomic analyses, we observed shorter serial intervals in case clusters and faster spread for Omicron relative to Delta. These findings highlight the importance of integrating genomic surveillance into university testing studies to better characterize VOC community spread.

Variants have continuously altered our understanding of SARS-CoV-2 genomic epidemiology. The adaptation of public health recommendations to this quickly changing landscape relies on rapid data collection, and university testing programs are uniquely positioned to collect data that may be more broadly representative of community dynamics than hospital-based surveillance strategies. Using symptom and exposure-based testing, we identified Omicron cases and characterized viral loads, serial intervals, and symptoms through daily online questionnaires in real-time as the first introductions of Omicron occurred. Prospective, longitudinal data collection from dormitories and other congregate settings offers an opportunity to understand the transmission dynamics of viral infections within clusters. For example, traditional household studies, including the Household Influenza Vaccine Evaluation[29] and the Seattle Virus Watch[30] have informed public health recommendations for influenza-related isolation and quarantine. Our university-based study with students residing in shared housing allowed for rapid data collection and decision-making around the evolving transmission dynamics of VOCs.

Our findings suggest a median serial interval of 2 days and 6 days among persons with Omicron and Delta, respectively. In contrast to other analyses examining serial intervals within households or other clusters[16–19,22], we used viral genomic data to minimize confounding of the serial infection interval by co-incident exposures during periods of high community transmission. By using only identical genomes to calculate the intraresidence serial interval, we decreased the likelihood that clusters are the result of more than one index case (although we cannot eliminate this possibility). Our finding of the reduced serial interval between the index and subsequent household infections for Omicron compared to Delta cases is consistent with other studies in

**Table 2 | Cycle threshold comparisons by Delta and Omicron variants**

| | Mean unadjusted difference in Orf1b Ct value (95% CI) | p-value[b] | Mean adjusted difference in Orf1b Ct value (95% CI)[a] | p-value[b] |
|---|---|---|---|---|
| **All Delta and Omicron positive individuals, adjusted for age, symptoms, and average RNase P gene value N = 1870 (Delta = 182, Omicron = 1688)** | | | | |
| Variant (Omicron vs. Delta) | 1.30 (0.76, 1.84) | **$2.5 \times 10^{-6}$** | 1.07 (0.58, 1.57) | **$2.5 \times 10^{-5}$** |
| Age (years) | −0.01 (−0.03, 0.01) | 0.29 | −0.01 (−0.03, 0.01) | 0.22 |
| Symptoms (symptomatic vs. asymptomatic) | −0.94 (−1.35, -0.53) | **$8.2 \times 10^{-6}$** | −1.11 (−1.50, −0.74) | **$9.3 \times 10^{-9}$** |
| Average RNase P gene value | 0.39 (0.35, 0.43) | **$2.3 \times 10^{-63}$** | 0.39 (0.35, 0.43) | **$9.3 \times 10^{-65}$** |
| **Symptomatic individuals (with symptom onset on or before day of swab) adjusted for age, days since symptom onset, and average RNase P gene value N = 1466 (Delta = 144, Omicron = 1322)** | | | | |
| Variant (Omicron vs. Delta) | 1.08 (0.48, 1.68) | **0.0004** | 0.81 (0.26, 1.37) | **0.004** |
| Age (years) | −0.01 (−0.03, 0.01) | 0.44 | −0.01 (−0.03, 0.01) | 0.29 |
| Days since symptom onset | 0.32 (0.22, 0.41) | **$1.8 \times 10^{-10}$** | 0.29 (0.20, 0.38) | **$1.6 \times 10^{-10}$** |
| Average RNase P gene value | 0.40 (0.35, 0.45) | **$3.2 \times 10^{-54}$** | 0.39 (0.34, 0.44) | **$1.2 \times 10^{-52}$** |
| **Vaccinated individuals (complete primary series or booster dose at time of swab), adjusted for age, symptoms status, vaccination status, days since last COVID-19 vaccine dose, and average RNase P gene value N = 1025 (Delta = 84, Omicron = 941)** | | | | |
| Variant (Omicron vs. Delta) | 1.26 (0.47, 2.05) | **0.001** | 0.87 (0.08, 1.67) | **0.03** |
| Age (years) | −0.003 (−0.03, 0.02) | 0.74 | −0.01 (−0.03, 0.01) | 0.48 |
| Symptoms (symptomatic vs. asymptomatic) | −1.27 (−2.86, −0.16) | **0.00001** | −1.41 (−1.93, −0.88) | **$1.6 \times 10^{-7}$** |
| Booster vaccination vs. Complete primary series vaccination | 0.71 (0.19, 1.22) | **0.007** | 0.50 (−0.55, 1.54) | 0.35 |
| Days since last COVID-19 vaccine dose | −0.002 (−0.005, −0.0004) | **0.02** | −0.001 (−0.006, 0.003) | 0.57 |
| Average RNase P gene value | 0.37 (0.31, 0.43) | **$7.8 \times 10^{-33}$** | 0.38 (0.32, 0.44) | **$3.8 \times 10^{-35}$** |

[a]Mean adjusted differences estimated using three multiple linear regression of average Orf1b Ct value on variant (Omicron vs. Delta) adjusted by covariates indicated in the table above. All regressions were restricted to Delta and Omicron cases detected using RHINOstic™ swabs (excludes 27 Delta and 42 Omicron cases detected using US Cotton #3 swabs).
[b]Two-sided t statistic with significance level of 0.05.
Bold values represent statistical significance $p < 0.05$.

the US (median serial interval of 3 days for Omicron)[16] and others in Europe and South Korea (reported mean serial intervals from 2.8 to 3.5 days for Omicron and 3 to 4.1 days for Delta)[19–22]. Our estimated median serial interval of 2 days for Omicron is lower than these studies, and this may be due to our study population being, on average, younger and more highly vaccinated and only one other study using genomic sequencing to identify household transmission[20].

We found that semiquantitative viral loads were lower for Omicron compared to Delta variant infections, supporting the theory that increased transmissibility of the Omicron variant is not due to viral load and in agreement with other studies in the setting of highly vaccinated populations, including other US universities, symptomatic healthcare workers in France, and the US National Basketball Association's (NBA) occupational health program[9,12]. In contrast, other studies did not find a difference in viral loads between Omicron and Delta variant infections, including in studies of hospitalized patients, symptomatic outpatients, and the general population in Portugal and Washington state[10,11]. The reasons for these differences in Ct values between Delta and Omicron variant infections are not clear. One difference in populations in the above-listed studies is age, yet we did not find an association between age and differences in Ct values between Delta and Omicron infections in this study. Ct value differences between populations could be due to unmeasured differences in these populations, or differences in testing practices between studies, with more asymptomatic or paucisymptomatic testing done on college campuses, healthcare workers, and NBA players in comparison to testing in the general population or in hospitalized patients.

The Omicron variant swiftly replaced Delta on campus, despite high rates of vaccination and broad campus mitigation measures in place. Due to the availability of rapid whole genome sequencing[31,32], we quickly identified the emergence of Omicron. The rapid rise of Omicron may have been facilitated by vaccine breakthrough cases and immune evasion associated with this variant, as reported in early Omicron studies[33–35]. Despite higher numbers of Omicron infections

after vaccination, early household transmission reports show that individuals who received a booster dose had lower secondary attack rates, lower risk of transmission, and fewer secondary infections[17]. The pace of Omicron variant spread in this population, quantified as an increased Rt compared to Delta variant, exemplifies that SARS-CoV-2 outbreaks may continue to occur despite stringent public health interventions. To mitigate further waves of SARS-CoV-2 transmission, community-based genomic surveillance studies should be leveraged to guide policy and containment strategies. This reality, and a needed shift in the national COVID-19 strategy to focus on a "new normal" in which risk reduction and hospital capacity are prioritized, are essential as we transition to the next phases of the pandemic[36].

Although SARS-CoV-2 variant classification may be achieved without full genome sequencing, the generation of complete viral genomes provided additional insight into viral transmission on campus. Our genomic data suggest that within 2 months of the first detection of Omicron on campus, there were at least 1000 distinct introductions of the variant, though our ability to precisely define the number of introduction events represented by the sequenced campus Omicron cases is limited and likely due to the limited genomic variation among Omicron viruses and the fact that study genomes currently make up about 10% of the available Omicron genomes from Washington. This estimate does suggest that most Omicron introduction events resulted in a single sequenced case. Our analyses indicate that the same is true for Delta introduction events. However, for Delta, it was also clear that most sequenced cases were the result of introduction events that resulted in multiple cases and that most on-campus SARS-CoV-2 cases due to Delta variant viruses were the result of campus-related transmission. It is particularly notable that most sequenced Delta cases were due to just one of three putative introduction events, while the highest number of cases due to a single putative Omicron introduction event (for the analysis including all Washington state Omicron sequences) was 41 (or 2.4% of the total number of sequenced cases), which may suggest differences in patterns of Delta and Omicron transmission on campus. Unfortunately,

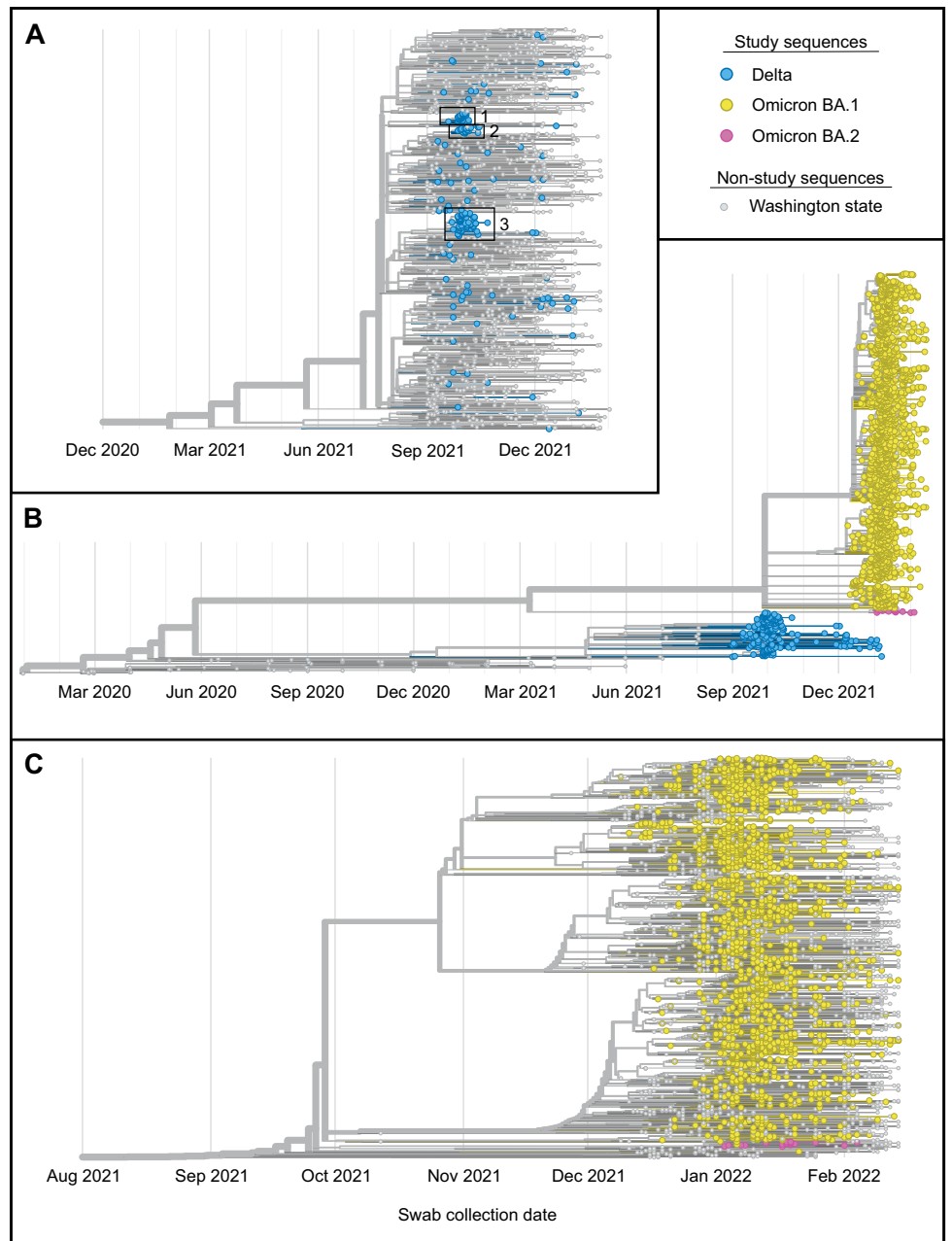

**Fig. 3 | Phylogenetics of sequenced campus viral genomes. A** Phylogenetic tree of 209 sequenced Delta samples collected on the UW campus and 1174 randomly selected genomes from samples collected in Washington during the same time period. Three monophyletic clusters containing exclusively or almost exclusively study genomes are boxed and numbered. **B** Phylogenetic tree containing sequences for campus samples collected between September 4, 2021 and February 14, 2022 (*N* = 1939) plus the Wuhan/Hu-1 reference genome and approximately 100 GISAID Washington state genomes collected from March 2020 to August 2021. The tree also contains genomes for 94 samples collected in Washington state from March 2020 to August 2021 (gray nodes) and the Wuhan/Hu-1 reference genome (gray node, far left) for context. Delta variant campus genomes are in yellow, and Omicron variant genomes are in blue. **C** Phylogenetic tree of 1730 sequenced Omicron samples collected on the UW campus plus the Wuhan-Hu-1 reference genome, and 1512 randomly selected genomes for samples collected in Washington during the same time period. Trees are available in the project Github repository: https://github.com/amcasto/huskytesting_deltaomicron.

the considerable degree of uncertainty in the Omicron phylogenetic tree limits our ability to directly compare transmission patterns of the two variants.

Our study limitations include the lack of routine surveillance testing of the entire campus population. Follow-up symptom data were missing for some individuals, and therefore we do not know if some asymptomatic cases were presymptomatic. We rely on self-report of vaccine status and could not reference state registries. However, state registry data may be incomplete or delayed, especially for students from other states. A limitation of our Ct analysis is the

change in swab type during the study, which may impact viral load, and we therefore restricted our viral load analysis to only one swab type. We also did not include repeat infections. Finally, this study included only people on a single university campus who participated in the research study and who are, on average, younger, healthier, and more educated than the general population.

In conclusion, we found the rapid replacement of the SARS-CoV-2 Delta variant with the Omicron variant within a highly vaccinated university population. As we move into the next phases of the pandemic, real-time data around viral kinetics and genomic epidemiology

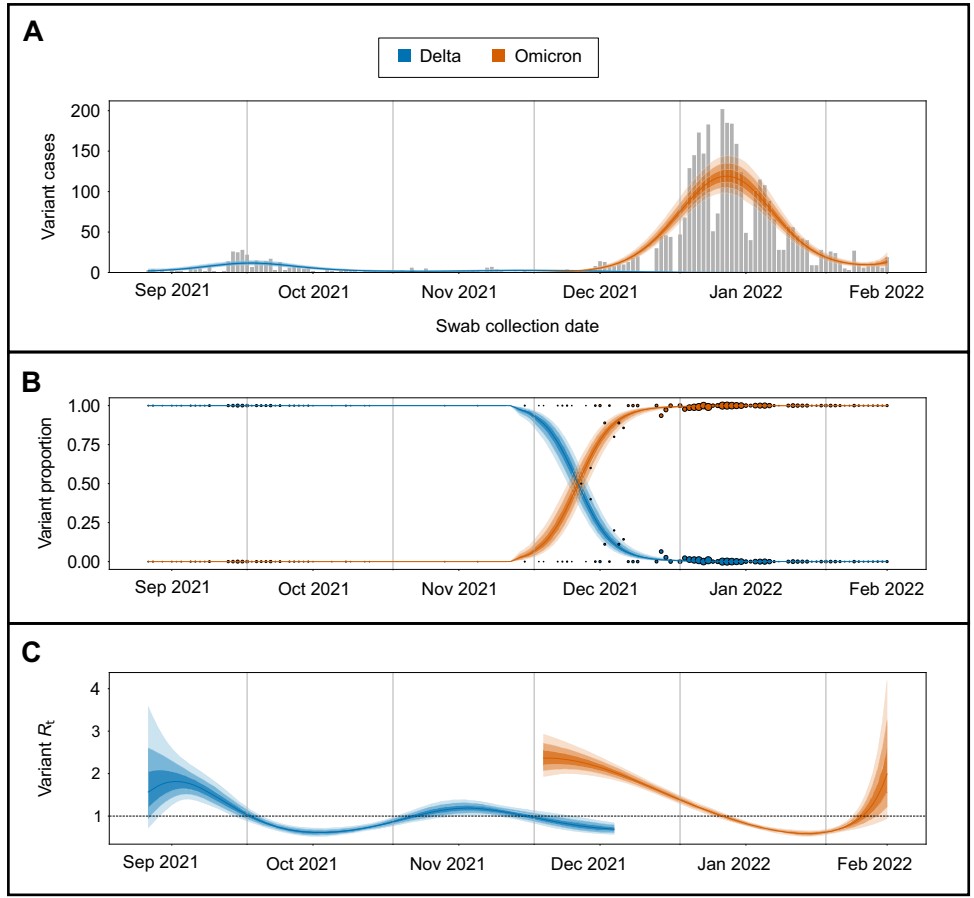

**Fig. 4 | Transmission dynamics of Delta and Omicron over study period.**
**A** SARS-CoV-2-positive samples (gray bars) against posterior variant-specific inci-dence over the study period. **B** Observed variant proportion of sequenced positive SARS-CoV-2 samples against posterior variant proportion. Radius of points corresponds to counts of sequenced samples for that day. **C** Posterior estimates of variant-specific effective reproduction numbers. Shaded intervals in all plots cor-respond to 50%, 80%, and 95% credible intervals.

of emerging variants will be important to guide our national strategies for mitigating respiratory virus spread.

## Methods

The Husky Coronavirus Testing (HCT) research study provides SARS-CoV-2 testing at the University of Washington (UW), a large public university in Seattle, Washington, USA[23]. University-wide mitigation policies did not vary during the course of the study and included an indoor mask mandate, improvements in air filtration, limitations in the size of gatherings, and mandatory vaccination for faculty, staff, and students, resulting in the completion of the primary vaccine series for 98.6% of students, 98.9% of staff, and 99.7% of faculty by January 2022[37]. Individuals were eligible to enroll in the study if they were faculty, staff, or students at the university and were English-speaking. Clinical symptoms and vaccination status were collected through electronic questionnaires. The race collected was self-reported, and the categories shown in Table 1 are identical to those presented in the questionnaire)[23]. Participants completed a daily attestation via email or text message, and those who reported new symptoms, exposure to a known SARS-CoV-2 case, or recent travel were offered SARS-CoV-2 testing. Additionally, participants could request testing for any reason. Data were collected using Project REDCap[38,39].

### Swab collection
Testing was performed through three mechanisms: observed self-swab at a staffed kiosk, unobserved self-swab returned to a campus testing dropbox, or unobserved self-swab returned to the laboratory via courier[40]. Two swab types were used; a US Cotton #3 swab (SteriPack Polyester Spun Swab), returned in a 10 mL tube, was used for all unobserved collection returned via courier and for some observed collection testing at times of supply chain issues. The RHINOsticTM Automated Nasal Swab (Rhinostics RH-S000001), returned in a MatrixTM 1.0 mL ScrewTop Tube (Thermo Fisher 3741), was used for observed kiosk and unobserved dropbox swab collections.

### Laboratory methods
All swabs were stored dry, with no preservative or media, and eluted with 1 mL Tris-EDTA for US Cotton #3, or 300 μL Tris-EDTA for RHI-NOsticTM. Fifty microliters of eluate was treated with proteinase K and heat for direct RT-qPCR (Swab-Express RT-qPCR) as previously described[41]. The RT-qPCR assay employs custom probe sets for SARS-CoV-2 Orf1b and S-gene designed against the ancestral strain that is multiplexed with a probe set for human RNase P[41]. Briefly, 5 uL of the prepared eluate was transferred to four multiplexed RT-qPCR reac-tions, two Orf1b-FAM plus RNase P-VIC and two S-FAM plus RNase P-VIC. Positive samples had SARS-CoV-2 targets detected in three or four reactions and an internal control RNase P amplification detected in at least three reactions; however, the 157–158 deletion in Delta var-iants results in S-gene target failure or delay in our assay.

### Genomic sequencing
Viral genome sequencing was attempted on SARS-CoV-2-positive specimens with a high quantity of SARS-CoV-2 RNA, generally having Orf1b cycle threshold (Ct) ≤30. Nucleic acids were extracted (Magna

Pure 96, Roche), and sequencing libraries prepared (Illumina COVID-Seq kit). Artic V4 primers were used starting November 18, 2021, to account for Beta and Delta spike protein variants (https://community.artic.network/t/sars-cov-2-version-4-scheme-release/312). The Artic V4.1 spike-in method was used starting January 12, 2022, to account for Omicron variant (https://community.artic.network/t/sars-cov-2-v4-1-update-for-omicron-variant/342). Genomes were sequenced (Illumina NextSeq2000 P200 kit), and consensus genomes were assembled against the SARS-CoV-2 reference genome Wuhan/Hu-1/2019 (Genbank accession MN908947, https://www.ncbi.nlm.nih.gov/nuccore/mn908947) using a modified iVar pipeline[42]. Consensus sequences were deposited to GenBank and GISAID (see Supplementary Materials). We considered "BA.1" to include the parental lineage and all BA.1 sublineages and "BA.2" to include the parental lineage and all BA.2 sublineages.

## Statistical analysis

We used the term "infection date" to describe the collection date of each person's first SARS-CoV-2-positive sample and to represent the first known date of infection regardless of symptom status. For participants who tested positive for SARS-CoV-2 more than once between September 10, 2021 and February 14, 2022, the first infection was included in our analysis. The proportion of cases reporting various symptoms was compared by variant using Pearson's chi-squared tests. The median serial interval of symptomatic participants in clusters was compared by variant using a nonparametric Mann–Whitney U test.

COVID-19 vaccination status was collected at enrollment, updated monthly, and updated at or after collection of the SARS-CoV-2-positive samples. Participants self-reported vaccine manufacturer name, dose number, and date of receipt. Vaccination status for participants is dynamic, and in this manuscript, the term "vaccination status" reflects the status on the date a positive swab was taken. Fully vaccinated was defined as completion of the primary series at least two weeks prior to the positive test date. Partially vaccinated was defined as an incomplete two-dose primary series or less than two weeks since the completion of the primary series. Unvaccinated was defined as confirmed no vaccination. Vaccination was defined as unknown for participants who reported invalid dates or no information at all. A participant was considered boosted if they received a booster dose at least two weeks prior to the positive test date, partially boosted if fewer than two weeks, and not boosted if no booster was received by the positive test date.

The shared residence was defined as the same apartment, dorm room, or unit number, or by the same street address for single-unit residences. Clusters of positive cases were defined as living within a shared residence with identical SARS-CoV-2 sequences. An index date and serial interval were calculated for each cluster with at least two symptomatic individuals. The serial interval was defined for each non-index individual in a cluster as the duration of time between the index symptom onset date to the non-index individual's symptom onset date. Symptom onset date was defined as the earliest symptom onset date within one week before or after each individual's positive test. Individuals were considered asymptomatic if they reported no symptoms within one week before and after testing positive.

Multiple linear regression models were used to estimate the mean difference in Ct between Omicron and Delta variant cases, adjusting for age, symptom status (symptomatic versus asymptomatic), average RNase P gene value, days since symptom onset among those with symptoms, and vaccination status (primary series vs. booster) and days since last COVID-19 vaccination among those fully vaccinated. Additionally, we estimated the mean difference in Ct between Omicron lineages BA.1 and BA.2. Regression analyses were restricted to RHINOstic_TM swabs due to previously observed differences in Ct between RHINOstic_TM swabs and US Cotton #3 swabs[23]. Mean Ct was calculated using only Orf1b Ct values due to differences in S-gene amplification

between Delta and Omicron. Analyses were conducted in R (R-4.1.1, R Core Team, 2021).

## Genomic/phylogenetic analyses

Genomic analyses included consensus genomes generated for this project (see above) and publicly available SARS-CoV-2 genomes for other Washington state samples from the GISAID EpiCoV database[43] (available in repository https://github.com/amcasto/huskytesting_deltaomicron). The latter sequences were screened using Nextclade version 1.10.0[44] and any sequences deemed to be of "bad" or "mediocre" quality by this tool (due to missing data, mixed sites, private mutations, mutation clusters, frameshifts, or stop codons) were excluded from further analyses.

Pairwise distances between genomes were calculated by summing the number of nucleotide differences between two genomes. Nucleotide positions with missing data (Ns) in either genome were not considered in the calculation of the pairwise distance between two genomes.

Maximum likelihood phylogenetic trees were constructed using the Nextstrain Augur software package[45] using default parameters for SARS-CoV-2 as outlined on the Nextstrain GitHub webpage[46]. In brief, the first 100 and final 50 nucleotides of SARS-CoV-2 genomes as well as positions 21,846 and 21,987 were masked before genomes were aligned using MAFFT v7.453. IQ-Tree v2.2.0 was then applied to alignments with the number of initial parsimony trees set to 10 and the number of search iterations set to 4. Trees were rooted on the reference SARS-CoV-2 genome, Hu/Wuhan-1/2019, and were generated using the GTR model. Timetree was used to estimate molecular clock branch lengths with the following parameters: fixed clock rate 0.0008, the standard deviation of the fixed clock rate 0.0004, and clock filter 8. Polytomies were resolved in cases where this could be done based on the collection dates of the various child nodes involved in a polytomy. Otherwise, polytomies were retained in the tree. Finally, Nextstrain Auspice software was used for tree visualization.

The phylogenetic tree shown in Fig. 3A includes the reference (Hu/Wuhan-1/2019) genome, 209 genomes generated from samples positive for Delta variant SARS-CoV-2 from the UW campus and 1174 GISAID genomes representing Delta variant SARS-CoV-2 samples collected in Washington. The latter 1174 were randomly selected from among 15,406 available GISAID genomes using the Nextstrain Augur filter command, with the maximum number of genomes per month for September 2021 to February 2022 capped at approximately 250. Similarly, the phylogenetic tree shown in Fig. 3C includes the reference (Hu/Wuhan-1/2019) genome, 1730 genomes generated from samples positive for Omicron variant SARS-CoV-2 from the UW campus, and 1512 GISAID genomes representing Omicron variant SARS-CoV-2 samples collected in Washington. The latter 1512 were randomly selected from among 15,406 available GISAID genomes using the Nextstrain Augur filter command with the maximum number of genomes per month for December 2021 to February 2022 capped at approximately 500. The tree in Fig. 3B includes the reference genome, 209 Delta genomes and 1730 Omicron genomes from the UW campus, and genomes corresponding to the first five (chronologically) samples collected in Washington each month for March 2020 to August 2021. Bootstrap values for the trees in Fig. 3A, C were calculated using IQ-Tree run with 1000 ultrafast bootstrap replicates.

To estimate the number of Delta viral introduction events onto campus represented by the 209 sequenced Delta samples, we created a tree that included these samples along with all GISAID Delta genomes from samples collected in Washington state between September 1, 2021, and February 14, 2022, meeting our quality criteria ($N = 15,406$). The Nextstrain Augur "traits" subcommand was used to infer campus versus non-campus states for all internal nodes. An introduction event

was presumed to have occurred in all cases in which a Washington state non-university campus parent node connected with an on-campus child node. To assess the accuracy of this estimate given the number of Washington state Delta genomes available, we recalculated the introduction event number using 15 subsamples of the total pool of available non-study genomes varying in size from $N = 1000$ to $N = 15,000$. For each of these subsamples, the entire process of calculating the number of introduction events was repeated, including tree construction and the assignment of states to internal nodes. We also estimated the number of introduction events of the Omicron variant onto campus represented by the 1730 sequenced cases using a tree that included 1730 sequenced Omicron samples and all GISAID Omicron samples that were collected in Washington state before February 14, 2022 and met quality criteria ($N = 14,359$) to obtain an estimate of the number of Omicron introduction events onto campus; the accuracy assessment for this estimate was done using 14 subsamples of Washington state Omicron genomes varying in size from $N = 1000$ to $N = 14,000$.

### Transmission dynamics

Using all sequenced study samples and overall daily case counts over the study period, we estimated variant-specific effective reproduction numbers (Rt) for Delta and Omicron. To do this, we reconstructed variant-specific incidence from observed daily variant proportions using a multinomial likelihood for variant proportion and negative binomial likelihood for cases. This reconstructed incidence was used to compute the effective reproduction number for Delta and Omicron while reflecting the observed shorter serial interval of Omicron versus Delta[28].

### Ethics statement

The UW IRB approved this study (#00011148), and all participants gave informed consent. Participants under 18 years of age also required consent from a parent or guardian.

### Reporting summary

Further information on research design is available in the Nature Research Reporting Summary linked to this article.

## Data availability

Genomic data are publicly available on GISAID and phylogenetic tree files are available on https://github.com/amcasto/huskytesting_deltaomicron. All genome sequences and associated metadata in this dataset are published in GISAID's EpiCoV database. To view the contributors of individual sequences with details, such as accession number, Virus name, Collection date, Originating Lab and Submitting Lab, and the list of Authors, visit https://doi.org/10.55876/gis8.220721zh. A list of the Genbank accession numbers is also shown in the Supplementary information. Nongenomic de-identified data are available by request to anaweil@uw.edu without a specified timeframe for requests.

## Code availability

Code used for the regression analysis is available in our supplemental files. All available data are de-identified. The GISAID Identifier is EPI_SET_20220721zh accessible at https://doi.org/10.55876/gis8.220721zh, and code used for genomic analysis is shown in https://github.com/amcasto/huskytesting_deltaomicron and is also available at https://zenodo.org/record/6903850#.YuIXw_HMKZw.

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

## Acknowledgements

We would like to thank the study participants. We also thank the UW Environmental Health & Safety COVID-19 Prevention & Response team, including Katia Harb, Sheryl Schwartz, Natalie Thiel, and Kim Baker, UW administration and Incident Command leadership group (Margaret Shepherd, Josh Gana, Pamela Schreiber), Chu Lab, the Brotman Baty Institute, Husky Coronavirus Testing team, Dr. Timothy Uyeki, Dr. Anna Wald, and Dr. Roy Burstein. This work was supported by the United States Senate and House of Representative Bill 748, Coronavirus Aid, Relief, and Economic Security Act. We have included in the Data Availability section information on the GISAID authors, laboratories, and contact information.

## Author contributions

Conceived of study: A.A.W., C.M.L., J.S., J.P.H., L.M.S., and H.Y.C. Designed study tools for data collection or analysis: A.M.C., J.O., P.D.H., G.S.G., Z.A., M.D.F., D.A.N., J.S., T.B., and L.M.S. Collected data or supervised data collection: P.D.H., L.G., E.M., M.T., Z.A., C.R.W., N.K.L., L.C.P., D.M., T.W., and K.M.M. Analyzed data or supervised analysis: A.A.W., K.G.L., A.M.C., J.C.B., J.O., A.M., M.D.F., T.B., J.P.H., and H.Y.C. Wrote manuscript: A.A.W., K.G.L., J.C.B., A.M.C., T.B., L.M.S., and H.Y.C. Edited manuscript: G.S.G., E.J.C., M.B., and J.A.E.

## Competing interests

H.Y.C. reports consulting with Ellume, Pfizer, The Bill and Melinda Gates Foundation, Glaxo Smith Kline, and Merck. H.Y.C. received research funding from Gates Ventures, Sanofi Pasteur, and support and reagents from Ellume and Cepheid outside of the submitted work. G.S.G. received research grants and research support from the US National Institutes of Health, the University of Washington, the Bill & Melinda Gates Foundation, Gilead Sciences, Alere Technologies, Merck & Co., Janssen Pharmaceutica, Cerus Corporation, ViiV Healthcare, Bristol-Myers Squibb, Roche Molecular Systems, Abbott Molecular Diagnostics, and THERA Technologies/TaiMed Biologics, Inc, all outside of the submitted work. J.A.E. reports research support from Gates Ventures, AstraZeneca, GlaxoSmithKline, Merck, and Pfizer, and consulting with Sanofi Pasteur, AstraZeneca, Teva Pharmaceuticals, and Meissa Vaccines, outside of the submitted work. M.B. reports research support from Vir Biotechnology, GSK, Regeneron, Gilead Sciences, Janssen Pharmaceutica, Ridgeback, Merck, Gates Ventures, and consulting with Vir Biotechnology, Moderna, Helocyte, and Merck outside of the submitted work. The remaining authors declare no competing interests.
