## [Peer Review File · Nature Communications]

Genomic surveillance of SARS-CoV-2 Omicron variants on a university campusREVIEWER COMMENTS

Reviewer #1 (Remarks to the Author):

This is a nice presentation of the comparative epidemiology and clinical characteristics of Delta and Omicron epidemics in a University using a relatively large cohort, and high testing and genome sequencing. Combining sequencing and clinical data and using now well-established methods the authors demonstrate greater transmission for Omicron, and with newer methods show shorter serial interval for Omicron than Delta. The manuscript is very written and the Figures and Tables are of high quality.

I only have a few minor comments for author's consideration.

1. It is not clear if control measures, including mandated mask-wearing or other measures, changed during the study period.
2. It would also be useful to incorporate any large changes in University population size over the study period (e.g. term holidays) that could affect the dynamics.
3. The clustering of Delta into three major clades, as opposed to Omicron is interesting, and I wonder if this could be explained by underlying changes in population behaviour.

Reviewer #2 (Remarks to the Author):

The manuscript by Weil et al. details the transmission dynamics and emergence of Omicron in a highly vaccinated university community. The authors take advantage of a rich source of clinical characteristics using data from September 2001 to February 2022 to compare the Omicron variant to the Delta variant and use phylogenetics to track the introduction of the variants into the community. This is a well written manuscript an interesting dataset with dense sampling of viral genomes paired with good clinical and epi data. While some of the analyses produced by the authors appear relatively robust enough to deliver the insightful dynamics of this specific outbreak. However, additional clarity is needed in many places throughout the manuscript and the absence of detail on the phylogenetic analyses needs to be addressed. Also, while this study does discuss the dynamics of transmission broadly in a university setting it fails to capture the same level of resolution and insights other published work from similar settings have (e.g. Aggarwal et al. Nat Commun 13, 751, 2022).

Specific comments:

1. Can the authors determine transmission dynamics of big clusters following their introduction. Do they have any level of detail on whether such large-scale dispersion events are related to specific social gatherings or university-wide events?
2. Many other papers have not reported a similar finding that Ct values were on average higher for Omicron cases vs. Delta. The authors briefly discuss this in the discussion, but I feel that these findings should be better contextualized to give the reader a better perspective about whether such findings are unique to this cohort or more generalizable.
3. Phylogenetic analyses are limited in detailed aside from default parameters as outlined on Nextstrain. The onus should not be on the reader to try and figure out how trees were reconstructed but the author. I would encourage the authors to be explicit as possible so this work can be re-created. For example, what model was used to reconstruct the trees?
4. The authors assume that the phylogenetic tree inferred is "correct". How have the authors dealt with the phylogenetic uncertainty of the reconstruction process and how were polytomies dealt with as this does complicate SARS-CoV-2 phylogenies. Did the authors consider bootstrapping as a way to measure the uncertainty?
5. Can the authors provide a public repository such as a Github page or similar where the inputs and outputs of nextstrain (e.g. json files and tree) can be housed
6. Can the authors list the GISAID IDs of those genomes used for phylogenetic reconstruction.
7. The rarefaction analyses that is performed to analyze the number of introductions is not clear. For each subsample are the authors reconstructing the phylogenetic tree each time (e.g. 15 times for Delta) or are they merely pruning the large tree to get to their required size? Again authors

should be more explicit here.

8. Line 241: "maximum pairwise distance between two study Delta samples was 60.." 60 what? 60 nucleotides/bases?

9. For figure 3C how were genomes from Washington state selected? It says random? Was diversity used at all to select these sequences i.e. select the most genetically similar sequences?

Weil manuscript NCOMMS-22-16988A Response to Review

REVIEWER COMMENTS

Reviewer #1 (Remarks to the Author):

This is a nice presentation of the comparative epidemiology and clinical characteristics of Delta and Omicron epidemics in a University using a relatively large cohort, and high testing and genome sequencing. Combining sequencing and clinical data and using now well-established methods the authors demonstrate greater transmission for Omicron, and with newer methods show shorter serial interval for Omicron than Delta. The manuscript is very written and the Figures and Tables are of high quality.

I only have a few minor comments for author's consideration.

1. It is not clear if control measures, including mandated mask-wearing or other measures, changed during the study period.

We have added to the manuscript additional information about control measures used on campus during the study period.

“University-wide mitigation policies did not vary during the course of the study, and included an indoor mask mandate, improvements in air filtration, limitations in the size of gatherings, and mandatory vaccination for faculty, staff, and students”.

2. It would also be useful to incorporate any large changes in University population size over the study period (e.g. term holidays) that could affect the dynamics.

We have added additional information about university population fluctuations to a revised Figure 1 to give additional context to our results.

3. The clustering of Delta into three major clades, as opposed to Omicron is interesting, and I wonder if this could be explained by underlying changes in population behaviour.

We have further analyzed the three Delta clusters to better understand the dynamics influencing these groups of infections. This information, and statistical testing for the purpose of understanding which demographic factors were differentially associated with these clusters, is shown in the new Supplemental Table 2.

Reviewer #2 (Remarks to the Author):

The manuscript by Weil et al. details the transmission dynamics and emergence of Omicron in a highly vaccinated university community. The authors take advantage of a rich source of clinical characteristics using data from September 2001 to February 2022 to compare the Omicron variant to the Delta variant and use phylogenetics to track the introduction of the variants into the community. This is a well written manuscript an interesting dataset with dense sampling of viral genomes paired with good clinical and epi data. While some of the analyses produced by the authors appear relatively robust enough to deliver the insightful dynamics of this specific outbreak. However, additional clarity is needed in many places throughout the manuscript and the absence of detail on the phylogenetic analyses needs to be addressed. Also, while this study does discuss the dynamics of transmission broadly in a university setting it fails to capture the same level of resolution and insights other published work from similar settings have (e.g.

Aggarwal et al. Nat Commun 13, 751, 2022).

Specific comments:

1. Can the authors determine transmission dynamics of big clusters following their introduction. Do they have any level of detail on whether such large-scale dispersion events are related to specific social gatherings or university-wide events?

To add additional context to our analysis, we have included information regarding major campus events, holidays, and periods of online and in person instruction, shown in our revised Figure 1. Calculated time of the most recent common ancestor of a group of sequences, such as any one of the Delta clusters, is inexact, so it is not possible to link the origin of any of the Delta clusters to a particular event. These clusters overlapped in time, mapping to early in the school year when there are dozens or more events daily, on and off campus, including Greek recruitment and residence hall move-in, each of which involves several thousand students. We felt it was speculative to attempt to identify discrete events associated with clusters using phylogenetic data alone (ie lacking GPS information on participant movement, for example, which could be used to support hypotheses that one of the clusters was associated with a particular event). For these reasons, we were not able to achieve any additional level of detail on mapping clusters to specific large-scale dispersion events. However, we have further analyzed the demographics of the Delta clusters and added a new Supplemental Table 2 to show the participant characteristics associated with each group of infections.

2. Many other papers have not reported a similar finding that Ct values were on average higher for Omicron cases vs. Delta. The authors briefly discuss this in the discussion, but I feel that these findings should be better contextualized to give the reader a better perspective about whether such findings are unique to this cohort or more generalizable.

We have reviewed updated literature published since this manuscript was first submitted for review. We found that higher Ct values in Omicron infections compared to Delta were also observed in several other highly vaccinated populations including several university campuses, French health care workers, and National Basketball Association players. This suggests that higher viral loads did not drive Omicron transmission in our study. The literature at this time does not suggest a specific host or demographic factor that determines Omicron compared to Delta Ct values on a population level. We have included below the studies we reviewed, which include variation in population demographics, symptom and vaccination status, and PCR target for SARS-CoV-2 testing. We have added to the discussion on this point in the manuscript, which reads:

“We found that semiquantitative viral loads were lower for Omicron compared to Delta variant infections, supporting the theory that increased transmissibility of the Omicron variant is not due to viral load and in agreement with other studies in the setting of highly vaccinated populations, including other US universities, symptomatic health care workers in France, and the US National Basketball Association’s (NBA) occupational health program^{9,12}. In contrast, other studies did not find a difference in viral loads between Omicron and Delta variant infections, including in studies of hospitalized patients, symptomatic outpatients, and the general population in Portugal and Washington state^{10,11}. Reasons for these differences in Ct values between Delta and Omicron variant infections is not clear. One difference in populations in the above listed studies is age, yet we did not find an association between age and differences in Ct values between Delta and Omicron infections in this study. Ct value differences between populations could be due to unmeasured differences in these populations, or differences in

testing practices between studies, with more asymptomatic or paucisymptomatic testing done in college campuses, health care workers, and NBA players, in comparison to testing in the general population or in hospitalized patients.”

Studies reviewed on this topic, newly referenced in the discussion:

Manuscript	Ct conclusions comparing Delta and Omicron VOCs
US university population, Weil et al (this study)	Omicron = higher Cts
Boston universities, Petros et al	Mixed results, Omicron = higher Cts at two universities, no difference detected at one university
Health care workers, France, Sentis et al	Omicron = higher Cts
US NBA, Hay et al	Omicron = higher Cts
Swiss symptomatic outpatients (only 18 Omicron cases), Puhach et al	No difference found
US hospital patients, Fall et al	No difference found
WA general population, Laitman et al	Mixed results, Omicron = higher Cts or no difference (depending on symptoms and testing platform)
Portugal general population, Kislaya et al	No difference

3. Phylogenetic analyses are limited in detailed aside from default parameters as outlined on Nextstrain. The onus should not be on the reader to try and figure out how trees were reconstructed but the author. I would encourage the authors to be explicit as possible so this work can be re-created. For example, what model was used to reconstruct the trees?

We agree that our methods should be more thoroughly described. Additional details about the methods utilized in the phylogenetic analyses, including the model used for tree construction, have been added to the methods section, and are shown here:

“In brief, the first 100 and final 50 nucleotides of SARS-CoV-2 genomes as well as positions 21846 and 21987 were masked before genomes were aligned using MAFFT v7.453. IQ-Tree v2.2.0 was then applied to alignments with the number of initial parsimony trees set to 10 and

the number of search iterations set to 4. Trees were rooted on the reference SARS-CoV-2 genome, Hu/Wuhan-1/2019, and were generated using the GTR model. Timetree was used to estimate molecular clock branch lengths with the following parameters: fixed clock rate 0.0008, standard deviation of the fixed clock rate 0.0004, clock filter 8. Polytomies were resolved in cases where this could be done based on the collection dates of the various child nodes involved in a polytomy. Otherwise, polytomies were retained in the tree.”

“The phylogenetic tree shown in Figure 3A includes the reference (Hu/Wuhan-1/2019) genome, 209 genomes generated from samples positive for Delta variant SARS-CoV-2 from the UW campus, and 1,174 GISAID genomes representing Delta variant SARS-CoV-2 samples collected in Washington. The latter 1,174 were randomly selected from among 15,406 available GISAID genomes using the Nextstrain Augur filter command with the maximum number of genomes per month for September 2021 to February 2022 capped at approximately 250. Similarly, the phylogenetic tree shown in Figure 3C includes the reference (Hu/Wuhan-1/2019) genome, 1,730 genomes generated from samples positive for Omicron variant SARS-CoV-2 from the UW campus, and 1,512 GISAID genomes representing Omicron variant SARS-CoV-2 samples collected in Washington. The latter 1,512 were randomly selected from among 15,406 available GISAID genomes using the Nextstrain Augur filter command with the maximum number of genomes per month for December 2021 to February 2022 capped at approximately 500. The tree in Figure 3B includes the reference genome, 209 Delta genomes and 1730 Omicron genomes from the UW campus, and genomes corresponding to the first five (chronologically) samples collected in Washington each month for March 2020 to August 2021. Bootstrap values for the trees in Figure 3A and 3C were calculated using IQ-Tree run with 1,000 ultrafast bootstrap replicates.”

4. The authors assume that the phylogenetic tree inferred is “correct”. How have the authors dealt with the phylogenetic uncertainty of the reconstruction process and how were polytomies dealt with as this does complicate SARS-CoV-2 phylogenies. Did the authors consider bootstrapping as a way to measure the uncertainty?

Polytomies were resolved in the trees included in our study in cases where this could be done based on the collection dates of the various child nodes involved in a polytomy (this is the default setting for the Nextstrain Augur software pipeline for SARS-CoV-2). Otherwise, polytomies were retained in the tree. To assist the reader in judging the amount of uncertainty inherent in the trees in Figures 3A and 3C, we have calculated bootstrap values for these trees. We have added to the results the average bootstrap value for both trees, and the values for each of the 3 clusters of study-derived Delta genomes observed in Figure 3A. We have added this information to the methods and results:

“Polytomies were resolved in cases where this could be done based on the collection dates of the various child nodes involved in a polytomy. Otherwise, polytomies were retained in the tree.”

“Bootstrap values for the trees in Figure 3A and 3C were calculated using IQ-Tree run with 1,000 ultrafast bootstrap replicates.”

“Bootstrap values for all 3 Delta clusters were 100%.”

“Bootstrap values were also on average much lower for nodes in the Omicron tree relative to the Delta tree (average bootstrap value 32.5% versus 66.8%).”

5. Can the authors provide a public repository such as a Github page or similar where the inputs and outputs of Nextstrain (e.g. json files and tree) can be housed

Files for the trees shown in Figure 3 have been posted to a project Github repository: https://github.com/amcasto/huskytesting_deltaomicron. This reference has been added to the manuscript.

6. Can the authors list the GISAID IDs of those genomes used for phylogenetic reconstruction.

We have included a list of the names and GISAID (EPI_ISL) IDs for the 1939 SARS-CoV-2 genomes generated for this project on the project Github repository (https://github.com/amcasto/huskytesting_deltaomicron). This has been added to the manuscript.

7. The rarefaction analyses that is performed to analyze the number of introductions is not clear. For each subsample are the authors reconstructing the phylogenetic tree each time (e.g. 15 times for Delta) or are they merely pruning the large tree to get to their required size? Again authors should be more explicit here.

In the analyses to assess the accuracy of the Delta and Omicron introduction number estimates, we constructed a new phylogenetic tree for each subsample of Washington state Delta (N = 15) and Omicron (N = 14) genomes. We have now included this detail in the methods section:

“To estimate the number of Delta viral introduction events onto campus represented by the 209 sequenced Delta samples, we created a tree that included these samples along with all GISAID Delta genomes from samples collected in Washington state between September 1, 2021, and February 14, 2022, meeting our quality criteria (N=15,406). The Nextstrain Augur “traits” subcommand was used to infer campus versus non-campus states for all internal nodes. An introduction event was presumed to have occurred in all cases in which a Washington state non-university campus parent node connected with an on-campus child node. To assess the accuracy of this estimate given the number of Washington state Delta genomes available, we re-calculated the introduction event number using 15 subsamples of the total pool of available non-study genomes varying in size from N=1,000 to N=15,000. For each of these subsamples, the entire process of calculating the number of introduction events was repeated, including tree construction and the assignment of states to internal nodes. We also estimated the number of introduction events of the Omicron variant onto campus represented by the 1,730 sequenced cases using a tree which included the 1,730 sequenced Omicron samples and all GISAID Omicron samples that were collected in Washington state before February 14, 2022 and met quality criteria (N=14,359) to obtain an estimate of the number of Omicron introduction events onto campus; the accuracy assessment for this estimate was done using 14 subsamples of Washington state Omicron genomes varying in size from N=1,000 to N=14,000.”

8. Line 241: “maximum pairwise distance between two study Delta samples was 60.” 60 what? 60 nucleotides/bases?

The pairwise distance measures the number of nucleotide differences/changes between two genomes (which may also be called the “Hamming Distance”). A description of how pairwise distances were calculated was added to the statement of the results and to the methods section:

“Pairwise distances between genomes were calculated by summing the number of nucleotide differences between two genomes. Nucleotide positions with missing data (Ns) in either genome were not considered in the calculation of the pairwise distance between two genomes.”

9. For figure 3C how were genomes from Washington state selected? It says random? Was diversity used at all to select these sequences i.e. select the most genetically similar sequences?

For Figure 3C, 1,512 genomes were randomly selected from among 15,406 available GISAID genomes from Washington State using the Nextstrain Augur filter command with the maximum number of subsampled genomes per month for December 2021 to February 2022 capped at approximately 500. We have added these details to the methods:

“The phylogenetic tree shown in Figure 3A includes the reference (Hu/Wuhan-1/2019) genome, 209 genomes generated from samples positive for Delta variant SARS-CoV-2 from the UW campus, and 1,174 GISAID genomes representing Delta variant SARS-CoV-2 samples collected in Washington. The latter 1,174 were randomly selected from among 15,406 available GISAID genomes using the Nextstrain Augur filter command with the maximum number of genomes per month for September 2021 to February 2022 capped at approximately 250. Similarly, the phylogenetic tree shown in Figure 3C includes the reference (Hu/Wuhan-1/2019) genome, 1,730 genomes generated from samples positive for Omicron variant SARS-CoV-2 from the UW campus, and 1,512 GISAID genomes representing Omicron variant SARS-CoV-2 samples collected in Washington. The latter 1,512 were randomly selected from among 15,406 available GISAID genomes using the Nextstrain Augur filter command with the maximum number of genomes per month for December 2021 to February 2022 capped at approximately 500. The tree in Figure 3B includes the reference genome, 209 Delta genomes and 1730 Omicron genomes from the UW campus, and genomes corresponding to the first five (chronologically) samples collected in Washington each month for March 2020 to August 2021.”

This revised manuscript also contains the following edits that were not requested by the reviewers:

1. Addition of middle initial of one author (Kathryn “N” McCaffery), which was omitted in the prior manuscript draft.
2. Revision of the in-image legend of Figure 2 to revise “Booster dose” to “Boosted” to portray our results more accurately.

REVIEWERS' COMMENTS

Reviewer #2 (Remarks to the Author):

The authors have addressed all my prior concerns and have added clarity throughout the text.